# Effects of moderate and high intensity isocaloric aerobic training upon microvascular reactivity and myocardial oxidative stress in rats

**Lorena Paes[1], Daniel Lima[2], Cristiane Matsuura[2], Maria das Graças de Souza[1], Fátima Cyrino[1], Carolina Barbosa[1], Fernanda Ferrão[1], Daniel Bottino[1], Eliete Bouskela[1], Paulo Farinatti[3,4] ***

**1** University of Rio de Janeiro State, Rio de Janeiro, Laboratory for Clinical and Experimental Research on Vascular Biology, Rio de Janeiro, Brazil, **2** University of Rio de Janeiro State, Rio de Janeiro, Department of Pharmacology and Psychobiology, Rio de Janeiro, Brazil, **3** University of Rio de Janeiro State, Rio de Janeiro, Laboratory of Physical Activity and Health Promotion, Rio de Janeiro, Brazil, **4** Salgado de Oliveira University, Niteroi, Graduate Program in Sciences of Physical Activity, Rio de Janeiro, Brazil

* paulofarinatti@labsau.org

**Data Availability Statement:** All relevant data are within the paper and its Supporting Information files.

## Abstract

Systemic and central cardiovascular adaptations may vary in response to chronic exercise performed with different intensities and volumes. This study compared the effects of aerobic training with different intensities but equivalent volume upon microvascular reactivity in cremaster muscle and myocardial biomarkers of oxidative stress in Wistar rats. After peak oxygen uptake ($VO_{2peak}$) assessment, rats (n = 24) were assigned into three groups: moderate-intensity exercise training (MI); high-intensity exercise training (HI); sedentary control (SC). Treadmill training occurred during 4 weeks, with exercise bouts matched by the energy expenditure (3.0–3.5 Kcal). Microvascular reactivity was assessed *in vivo* by intravital microscopy in cremaster muscle arterioles, while biomarkers of oxidative stress and eNOS expression were quantified at left ventricle and at aorta, respectively. Similar increasing *vs.* sedentary control group (SC) occurred in moderate intensity training group (MI) and high-intensity training group (HI) for endothelium-dependent vasodilation ($10^{-4}$M: MI: 168.7%, HI: 164.6% *vs.* SC: 146.6%, *P* = 0.0004). Superoxide dismutase (SOD) (HI: 0.13 U/mg *vs.* MI: 0.09 U/mg and SC: 0.06 U/mg; *P* = 0.02), glutathione peroxidase (GPX) (HI: 0.00038 U/mg *vs.* MI: 0.00034 U/mg and SC: 0.00024 U/mg; *P* = 0.04), and carbonyl protein content (HI: 0.04 U/mg *vs.* MI: 0.03 U/mg and SC: 0.01 U/mg; *P* = 0.003) increased only in HI. No difference across groups was detected for catalase (CAT) (*P* = 0.12), Thiobarbituric acid reactive substances (TBARS) (*P* = 0.38) or eNOS expression in aorta (*P* = 0.44). In conclusion, higher exercise intensity induced greater improvements in myocardium antioxidant defenses, while gains in microvascular reactivity appeared to rely more on exercise volume than intensity.

**Funding:** The authors received no specific funding for this work.

**Competing interests:** The authors have declared that no competing interests exist.

**Abbreviations:** A.U., arbitrary units; eNOS, endothelial nitric oxide synthase; SC, sedentary control group; MI, moderate-intensity exercise training group; HI, high-intensity training exercise group; SOD, superoxide dismutase enzyme; CAT, catalase enzyme; GPX, glutathione peroxidase enzyme; ip, intraperitoneal; TBARS, thiobarbituric acid reactive substances; ROS, reactive oxygen species; EE, energy expenditure; $VO_2$, oxygen uptake; $VO_2R$, oxygen uptake reserve; EPOC, excess post-exercise oxygen consumption; ACh, acetylcholine; SNP, sodium nitroprusside; MDA, Malondialdehyde; DNPH, dinitrophenylhydrazine; SEM, standard error of mean.

## Introduction

Aerobic training is widely acknowledged as an effective strategy to maintain health and reduce cardiovascular risk [1]. Microvascular endothelial function [2–5] and myocardial antioxidant defenses [6–9] have been extensively investigated within this context, as reflecting early systemic and central cardiovascular changes [2, 6, 10, 11]. In regards to endothelial function, albeit chronic exercise seems to improve vasodilation of microvessels due to shear stress and circulating factors [2, 12], some research suggested that too vigorous training may increase oxidative stress and inflammation, therefore leading to deterioration of endothelial function [13, 14]. On the other hand, myocardium integrity seems to be favored by high-intensity training [15], due to greater production of antioxidants protecting against reactive oxygen species (ROS) [7, 16]. In short, different exercise intensities may elicit dissimilar chronic effects upon systemic and central cardiovascular markers–although endothelial function may be jeopardized by high-intensity training [14, 17], myocardium antioxidant protection could be benefited [15].

Thus, for a better understanding of cardiovascular benefits and risks due to exercise training, attention must be given to the relative role of its intensity and volume. A possible approach to address this question would be to compare the effects of aerobic training performed with different intensities, but similar overall volume as defined by the energy expenditure (EE)–in other words, isocaloric training bouts. Given that improvements in endothelial function may rely on the amount of EE during exercise [17, 18], it is feasible to speculate that isocaloric protocols would be able to induce favorable effects regardless of training intensity [19]. Moreover, this approach would help to avoid bias, since protocols with higher intensities can also be related to greater EE. This confounding factor precludes isolating the specific effects of exercise intensity upon endothelial function or myocardium integrity.

To date no study using animal models investigated the relative effects of exercise intensity and volume on endothelial function of microvessels (systemic cardiovascular marker), and antioxidant protection in myocardium (central cardiovascular marker). Thus, the present study aimed to investigate the effects of aerobic training performed with different intensities but equivalent volume, on microvascular reactivity in striated muscle and biomarkers of oxidative stress in myocardium of Wistar rats.

## Methods

### Ethical approval

Twenty-four male Wistar rats (*Rattus norvegicus*, Anilab, RJ, Brazil) were kept under 12:12-hour light-dark cycle in a temperature-controlled environment (22°C) with free access to water and standard rat chow (Nuvilab CR-1, Nuvital™, Curitiba, PR, Brazil). Experiments were performed according to principles of laboratory animal care (NIH pub. No. 86–23, revised 1996) and the protocol was approved by the Ethical Committee of the University of Rio de Janeiro State (License number: 024/2015).

### Study design

After assessment of oxygen uptake at rest ($VO_{2rest}$) and maximal exercise ($VO_{2peak}$), the animals (270 g, 12 weeks old) were randomly assigned into three groups: (a) moderate-intensity exercise training (MI; n = 8); (b) high-intensity exercise training (HI; n = 8) and (c) sedentary control (SC; n = 8). Two isocaloric exercise bouts were performed after the maximal exercise testing, in order to match the duration of training sessions according to the overall EE. After this, HI and MI underwent exercise bouts during four weeks on a motorized treadmill. At the

end of the training period, cardiorespiratory fitness and microvascular reactivity were assessed *in vivo*. At the end of each microcirculation experiment, animals profoundly anesthetized were euthanized by exsanguination. The eNOS expression was analyzed from aorta fragments. Left ventricle fragments were collected and immediately frozen in liquid nitrogen for measuring biomarkers of oxidative stress (antioxidants and oxidized biomolecules).

## Maximal graded exercise testing

Oxygen uptake at rest ($VO_{2rest}$) and during maximal exercise ($VO_{2peak}$) were determined by indirect calorimetry via metabolic cart (Oxylet[TM], Panlab Harvard Apparatus, Barcelona, Spain). The gas analyzer was coupled to a treadmill in a Plexiglas chamber, connected through a tube to an air pump used to maintain the airflow inside the chamber. The gas analyzer continuously measured relative concentrations of oxygen ($O_2$) and carbon dioxide ($CO_2$) effluent in the chamber. The $VO_2$ was calculated by specific software (Metabolism[TM], Panlab Harvard Apparatus, Barcelona, Spain) using equations described elsewhere [20]. Standard conditions of temperature, pressure and humidity (*STPD*) were kept in all experiments.

The $VO_2$rest was assessed during 30 min and data obtained during the last 5 min were averaged and recorded. Prior to $VO_{2peak}$ assessment, the rats underwent treadmill adaptation sessions during three days, with speed set at 16 cm/s during 10–15 min. The testing protocol consisted of load increments of 8 cm/s every 3 min, until the rats were no longer able to run. Exhaustion was determined when animals remained at the end of the treadmill (electrical shock grid) for 5 seconds and $VO_{2peak}$ corresponded to the highest $VO_2$ obtained during the test [20]. The shock grid delivered very low electrical currents (0.2 mA) and was used only to induce the rats to run.

## Training protocol (isocaloric exercise bouts)

The target workload during exercise bouts was calculated using the oxygen uptake reserve ($VO_2R$) method, as previously described [19]: $VO_2R$ = (fraction intensity) ($VO_{2peak}-VO_{2rest}$) + $VO_{2rest}$, where $VO_{2peak}$ corresponded to the highest $VO_2$ during maximal exercise testing. The relative intensity was defined according to each group; animals assigned to MI and HI exercised at speeds corresponding to 50% and 80% of $VO_2R$, respectively. Running speeds matching the relative intensities were individually calculated, based on $VO_2$ obtained in the maximal exercise testing.

The duration of isocaloric bouts was calculated (predicted) from values of $VO_2R$ and then converted to EE, as described elsewhere [20]. In order to confirm that both MI and HI bouts elicited similar EE, the $VO_2$ was measured throughout two exercise bouts (test and retest). When it was necessary, adjustments in predicted duration were made before the second bout, to ensure that the rats would perform the isocaloric exercise sessions with different intensities. Since the EE equivalence was confirmed in the retest, HI and MI groups performed the isocaloric training sessions without indirect calorimetry measurements. The exercise sessions were performed on a motorized treadmill (Insight Scientific Equipments[TM], São Paulo, SP, Brazil) during four weeks, five times a week, according to individual values of duration and running speed. In order to assess the excess post-exercise oxygen consumption (EPOC), the $VO_2$ was assessed during 30 min following the isocaloric bouts in a randomized subgroup of four animals selected from HI and MI.

## Assessment of microvascular reactivity

A standardized surgical procedure to evaluate microvascular reactivity in cremaster muscle was performed [21, 22]. In brief, the rats were anesthetized with ketamine and xylazine ip (65

and 10 mg·kg$^{-1}$ respectively), and the connective tissue was separated of cremaster. The muscle was then exposed on glass stage surface by pins fixed in edges of tissue. The cremaster was continuously superfused at a rate of 4 ml/min by HEPES-supported $HCO_3^-$ buffered saline solution [composition in mM: NaCl 110.0, KCl 4.7, $CaCl_2$ 2.0, $MgSO_4$ 1.2, $NaHCO_3$ 18.0, N-2-hydroxyethylpiperazine-N′-2ethanesulfonic acid (HEPES) 15.39 and HEPES Na$^+$-salt 14.61] bubbled with 5% $CO_2$−95% $N_2$. The pH was set at 7.4 and the temperature of superfusion solution was maintained at 37.5°C. The preparation was placed under an intravital microscope (Leica$^{TM}$ DMLFS, optical magnification ×600, NA 0.65, Wetzlar, Germany) coupled to a closed-circuit TV system, in order to record images of the arterioles. The cremaster preparation was maintained during 30 min at rest, before starting the experimental protocol to evaluate microvascular reactivity, as described elsewhere [22, 23].

Three arterioles (2$^{nd}$ and 3$^{rd}$ order) were analyzed in each cremaster preparation. After 30 min of rest, the images of arterioles were collected at baseline and after topical application of acetylcholine (ACh) and sodium nitroprusside (SNP) (Sigma-Aldrich$^{TM}$, St. Louis, MO, USA) at $10^{-8}$, $10^{-6}$ and $10^{-4}$ M. ACh and SNP were used to evaluate endothelium dependent and independent vasodilation, respectively. Each application was performed during 10 min, with a syringe infusion pump (model 55–2222, Harvard Apparatus$^{TM}$, Boston, MA, USA), producing a cumulative dose–response curve. Internal diameter of arterioles in each moment was measured by specific software (Image J $^{TM}$, U.S. NIH, Bethesda, MD, USA). At the end of each experiment, rats were sacrificed by anesthetic overdose followed by exsanguination, by removal of myocardium and aorta.

## Assessment of eNOS expression (Western blot analysis)

The thoracic aorta was dissected and protein extracted, as previously described [23]. In brief, the aorta was lysed in 50 mM HEPES (pH 6.4), 1% Triton X-100, 1 mM MgCl2, 10 mM EDTA, 1 mg/ml DNase, 0.5 mg/ml RNase containing the following protease inhibitors: 1 mM benzamidine, 1 mM PMSF, 1 mM leupeptin, and 1 mM soybean trypsin inhibitor (Sigma-Aldrich$^{TM}$, St. Louis, MO, USA). Protein content was measured using Pierce$^{TM}$ BCA Protein Assay Kit (Thermo Fisher Scientific$^{TM}$, MA, USA) and samples containing 50 μg of protein were resolved by electrophoresis (7.5% SDS-PAGE), transferred to PVDF membranes and stained with Ponceau to verify whether the same quantity of protein is present in all lanes. Proteins in PVDF membranes were probed with mouse monoclonal anti-eNOS (1:1000; Becton Dickinson, NJ, USA), incubated overnight at 4°C. A Ponceau Red staining was used as loading control. After extensive washings in TBS-Tween, PVDF membranes were incubated for 2 h at room temperature with horseradish peroxidase-conjugated secondary antibody anti-mouse IgG, diluted 1:5,000 and developed using Amersham ECL Western Blotting Detection Kit system (GE Healthcare Life Sciences$^{TM}$, Pittsburgh, PA, USA).

## Enzymatic assays and assessment of oxidative damage

Left ventricular tissue was dissected and homogenized (about 200 mg of tissue) on ice in PBS buffer 0.1 M (0.1 M NaCl, 0.1 M $NaH_2PO_4.H2O$, 0.1 M $NaH_2PO_4.2H_2O$, 0.1 M KCl, 6 mM EDTA, pH 7.5). Samples were centrifuged at 5,000 rpm for 20 min at 4°C and supernatant was collected. All samples were stored at -80°C for further analysis of enzymatic assays and oxidative damage. SOD, GPX and CAT activity were evaluated in left ventricle homogenate. Results are expressed as U/g of protein. Total protein content was quantified using the BCA assay kit (Bioagency$^{TM}$, Sao Paulo, SP, Brazil).

Measurement of SOD activity is based on its inhibition by pyrogallol autoxidation and assessed by spectrophotometric readings at 420 nm during 5 min [24, 25]. Catalase activity

was assessed by standard methods, as described elsewhere [26]. Briefly, this method is based on the rate of hydrogen peroxide decomposition, following the decay in absorbance at 240 nm during 1 minute. GPx activity was assessed by the rate of NADPH disappearance, measured by spectrophotometry (340 nm, during 3 min reading) [27].

The oxidative damage of proteins was assessed through formation of carbonyl groups based on the reaction with dinitrophenylhydrazine (DNPH) [28]. Carbonyl contents were determined by spectrophotometry at 370 nm. Lipid membrane damage was quantified by the formation of byproducts of lipid peroxidation (malondialdehyde, MDA), which are thiobarbituric acid reactive substances (TBARS). The MDA reacts with thiobarbituric acid resulting in a pinkish substance, which is subsequently analyzed by spectrophotometry [29]. TBARS were determined by reading the absorbance at 532 nm (Fluostar Omega$^{TM}$, BMG Labtech, Ortenberg, Germany).

### Statistical analysis

Normal distribution was ratified by the Kolmogorov-Smirnov test for data regarding maximal graded tests and isocaloric exercise bouts, which are presented as mean ± SEM. Microvascular reactivity are presented as median ($1^{st}$ – $3^{rd}$ quartile). Comparisons between pre *vs*. post-exercise training were performed for $VO_2$ and maximal speed by means of 2-way repeated measures ANOVA, followed by LSD post hoc testing in the event of significant *F* ratios. Microvascular reactivity and biomarkers of oxidative stress were compared only at post training, using Kruskal-Wallis test followed by Dunn test as post hoc verification. In all cases significant level was set at $P \leq 0.05$ and calculations were performed using the Statistica 10.0 software (Statsoft$^{TM}$, Tulsa, OK, USA).

### Results

Table 1 exhibits data of cardiorespiratory fitness assessed by maximal exercise testing, before and after training. At baseline, $VO_{2peak}$ ($P = 0.98$) and maximal running speed ($P = 0.38$) were similar across groups. After training, $VO_{2peak}$ decreased in SC ($P = 0.007$), increased in HI ($P = 0.001$) and increased twice in HI than MI, although this difference lacked of statistical significance ($VO_{2peak}$ Δ = 4.9 *vs*. 2.2 ml.kg.$^{-1}$min.$^{-1}$; $P = 0.12$). On the other hand, the increase in maximal speed was significantly greater in HI *vs*. MI ($P = 0.016$) and *vs*. SC ($P = 0.02$). Despite the differences detected for $VO_2$ after training, gains in body mass were similar across groups during all the experimental period ($P = 0.59$) (Fig 1), as well as the energy intake (SC: 94.9 ± 2.2 kcal/day; MI: 93.3 ± 4.8 kcal/day; HI: 92.9 ± 4.2 kcal/day; $P = 0.58$).

Table 1 also depicts data extracted from the first and second isocaloric bouts (test and retest sessions). As expected, running speeds ($P = 0.013$) and therefore target $VO_2$ ($P = 0.0002$) were always higher in HI *vs*. MI. In the first exercise bout, total EE was significantly different ($P = 0.007$) and could not be matched between groups, because animals in HI were not able to complete the predicted exercise duration before exhaustion. The second bout was performed after adjustments (by reducing duration for MI group) and differences in total EE between groups were no longer detected ($P = 0.61$), albeit target $VO_2$ in HI has remained higher than MI ($P = 0.007$). This is reinforced by the EPOC, which was significantly influenced by exercise intensity. At the beginning of recovery, HI had significantly higher $VO_2$ than MI (HI = 17.58 ± 1.4 mL/min *vs*. MI = 13.15 ± 0.8 mL/min vs.; $P = 0.03$) and this pattern was extended until the end of recovery, as shown by the $VO_2$ range (HI = 7.85 ± 0.6 mL/min *vs*. MI = 4.45 ± 0.8 mL/min; $P = 0.01$).

Changes in arterioles diameter in relation to basal conditions (considered as 100%) under topical application of ACh and SNP are depicted in Fig 2. Baseline refers to a period before

**Table 1. Data from maximal graded exercise test and isocaloric exercise training protocol.**

| *Maximal Graded Test* | | SC | HI | MI |
|---|---|---|---|---|
| **VO$_2$ peak (ml.kg.$^{-1}$min.$^{-1}$)** | Before | 78.3 (1.9) | 78.5 (2.5) | 77.9 (2.7) |
| | After | 74.5 (1.6) [A] | 83.4 (3.2) [A] | 80.1 (2.4) |
| | Δ | -3.7 (1.3) | 4.9 (1.0) [B] | 2.2 (1.1) [B] |
| **Maximal speed (cm.s$^{-1}$)** | Before | 66.0 (2.0) | 61.0 (2.1) | 65.0 (3.5) |
| | After | 61.0 (2.5) | 74.0 (2.9) [A B] | 68.0 (3.7) |
| | Δ | -5.0 (2.1) | 13.0 (3.0) [* B] | 3.0 (2.1) [B] |
| *Isocaloric bouts (predicted)* | | SC | HI | MI |
| **Target VO$_2$ (ml.min$^{-1}$)** | | - | 20.4 ±0.5 [*] | 14.1 ±0.7 |
| **Total EE (kcal)** | | - | 5.0 | 5.0 |
| **Duration (min)** | | - | 52.6 ±2.6 [*] | 63.6 ±2.3 |
| **Running speed (cm.s$^{-1}$)** | | - | 44 ±2 [*] | 18 ±1 |
| *Isocaloric bouts (1$^{st}$ session)* | | SC | HI | MI |
| **Target VO$_2$ (ml.min$^{-1}$)** | | - | 18.5 ±0.7 [*] | 14.5 ±0.5 |
| **Total EE (kcal)** | | - | 2.7 ±0.2 [* #] | 4.2 ±0.1 [#] |
| **Duration (min)** | | - | 31.6 ± 1.4[* #] | 57.4 ±1.5 |
| **Running speed (cm.s$^{-1}$)** | | - | 44 ±2 [*] | 18 ±1 |
| *Isocaloric bouts (2$^{nd}$ session)* | | SC | HI | MI |
| **Target VO$_2$ (ml.min$^{-1}$)** | | - | 18.5 ±0.5 [* #] | 14.6 ±0.7 |
| **Total EE (kcal)** | | - | 3.3 ±0.1 [#] | 3.4 ±0.1 [#] |
| **Duration (min)** | | - | 38.4 ±1.2 [* #] | 48.3 ±1.8 [#] |
| **Running speed (cm.s$^{-1}$)** | | - | 44 ±2 [*] | 18 ±1 |

Data are showed as mean ± SEM (n = 24).

Δ: Absolute value obtained after training minus before training

[A]: significant difference *vs.* before training

[B]: significant difference *vs.* SC group

[*]: significant difference *vs.* MI group

[#]: significant difference *vs.* own predicted values.

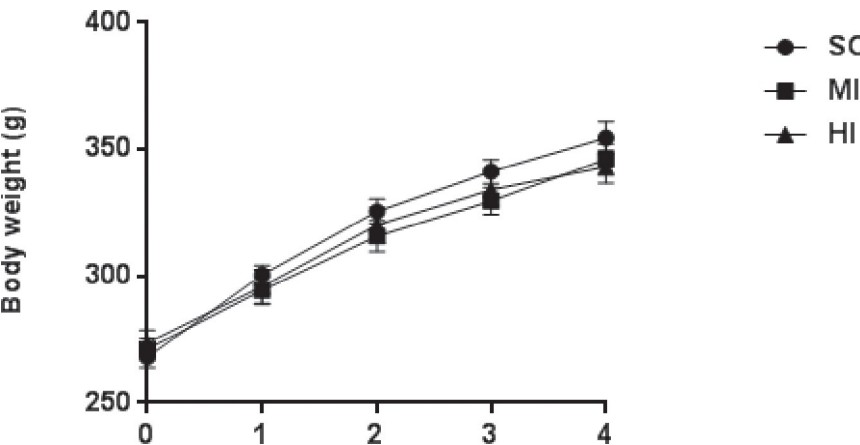

**Fig 1. Body mass during the experimental period (4 weeks).** Data are expressed as mean ± SEM (n = 24). No significant difference was found between the groups (*P* = 0.59).

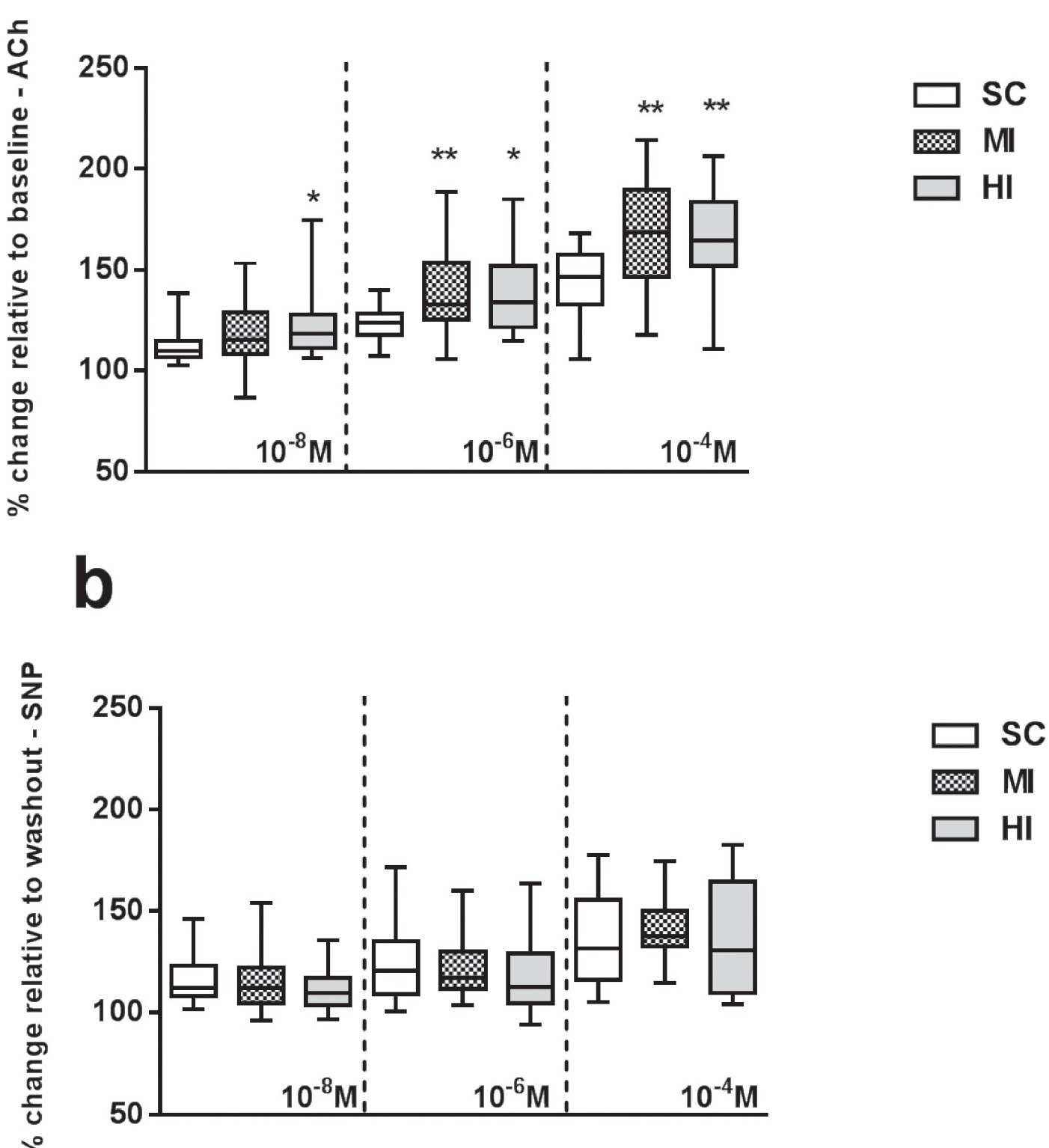

**Fig 2. Microvascular reactivity in vivo of arterioles in the cremaster muscle at the end of the experimental period (4$^{th}$ week).** Data are expressed as median (1$^{st}$ – 3$^{rd}$ quartile) (n = 24). (a) Endothelium-dependent vasodilation; (b) endothelium-independent vasodilation. *: Significant difference *vs*. SC (P < 0.05).

ACh or SNP application and after 30-min accommodation; at this phase, mean diameter was always similar across groups (SC: 71.73 ± 2.9 μm, MI: 74.91 ± 3.0 μm, HI: 70.55 ± 4.9 μm; $P = 0.54$). Endothelium-dependent vasodilation in response to the highest ACh concentration ($10^{-4}$M) and to the intermediate concentration ($10^{-6}$M) was significantly lower in SC than MI ($P = 0.0016$ and $P = 0.003$) and HI ($P = 0.0012$ and $P = 0.01$). In response to the lowest concentration of ACh ($10^{-8}$M), HI presented significantly increased vasodilation *vs*. SC ($P = 0.02$), but not MI ($P = 0.17$). No difference among groups was detected regarding endothelium-independent vasodilation induced by SNP at any concentrations ($10^{-8}$M: $P = 0.34$; $10^{-6}$M: $P = 0.49$; $10^{-4}$: $P = 0.28$).

Exercise performed with different intensities elicited different adaptations in antioxidant enzymatic activity and oxidative damage at left ventricle homogenates, as shown in Fig 3. CAT activity did not change in any group ($P = 0.12$). In contrast, SOD and GPX activity were significantly increased in HI ($P = 0.04$; $P = 0.01$), but not MI ($P = 0.29$; $P = 0.26$). Protein carbonyl content was higher in HI ($P = 0.003$), suggesting an increase in protein oxidative damage, whereas no change was observed after training in MI ($P = 0.07$). Malondialdehyde content did not differ among groups ($P = 0.38$). Fig 4 exhibits results of eNOS expression in aorta. Western Blot analysis could not detect differences between groups (HI = 1.10 ± 0.2 a.u., MI = 1.39 ± 0.2 a.u., SC = 1.00 ± 0.1 a.u.; P = 0.44).

## Discussion

This study investigated the effects of high- and moderate-intensity aerobic training with equivalent volume (or 'dose' reflected by EE) upon microvascular reactivity in cremaster muscle and biomarkers of oxidative stress in myocardium, considered as central and systemic cardiovascular health markers, respectively. It has been hypothesized that protocols with equivalent training volume would elicit similar outcomes, regardless of differences in training intensity. At least for microcirculation, our findings confirmed this hypothesis, since improvements in endothelium-dependent vasodilation were similar across training groups. On the other hand, only high-intensity training was able to improve SOD, GPX, and protein carbonyl content in myocardium.

Some prior research investigated the effects of different exercise intensities upon isolate vasculature [2, 13, 14] or myocardium [9, 15]. However, we could not find studies assessing the concomitant effects of aerobic training upon those peripheral and central markers. Evidently, this approach seems to be more adequate to investigate whether adaptations induced by different training protocols upon a given marker would extend to others. Moreover, bias related to exercise volume has never been addressed by previous research about effects of exercise intensity upon different cardiovascular markers [14, 30, 31].

Consistently with our initial hypothesis, significant improvements in endothelium-dependent vasodilation were similar across training groups in the highest concentration of ACh (MI = 168.7% *vs*. HI = 164.6%; $P = 0.91$), suggesting that effects upon microcirculation would be more related to exercise volume than intensity. This finding concurs with data previously reported, suggesting that potential effects of chronic exercise on vasculature would be rather associated to overall exercise 'dose' (as reflected by EE) than isolate relative intensity or duration [17, 18]. Potential mechanisms underlying chronic adaptations in endothelium include increased shear stress and NO bioavailability [2, 32]. Another finding that reinforces the beneficial role of these mechanisms on the endothelium function is the lack of differences in

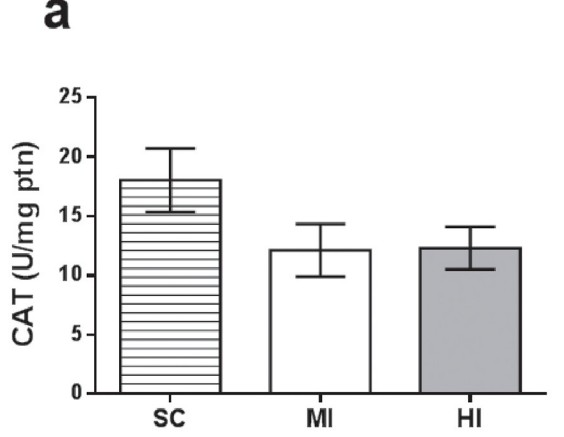

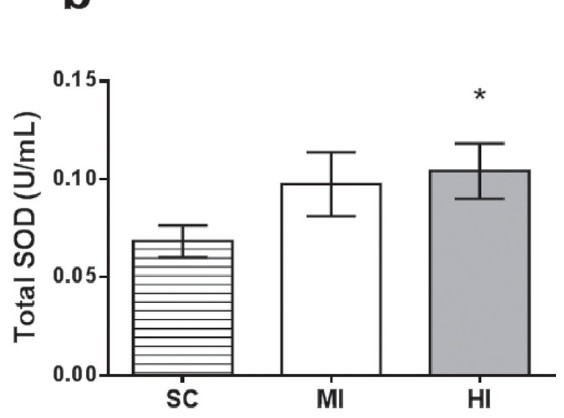

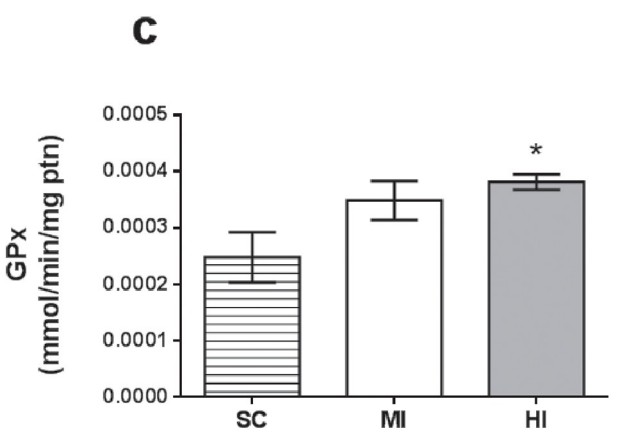

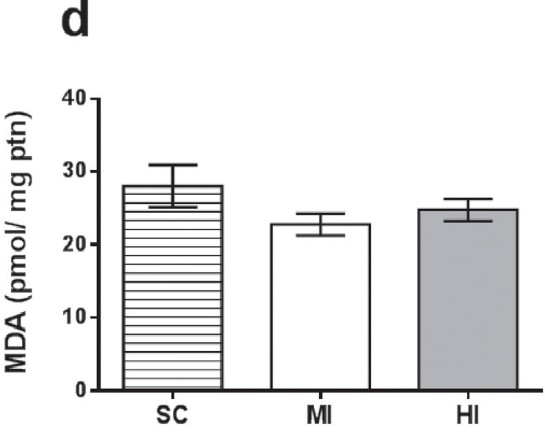

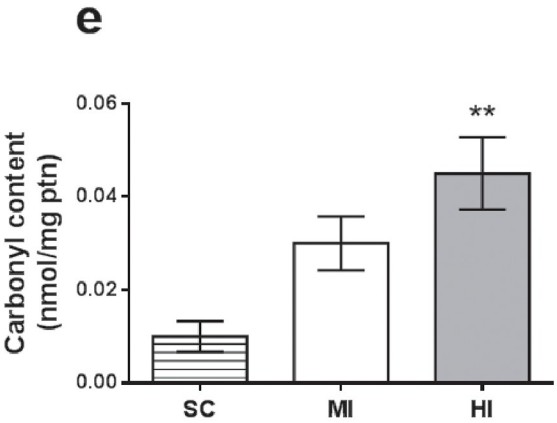

**Fig 3. Biomarkers of oxidative stress analyzed on left ventricle samples at the end of the experimental period (4ᵗʰ week).** Data are expressed as mean ± SEM (n = 24). (a) CAT Activity; (b) SOD Activity; (c) GPX Activity; (d) TBARS—MDA content; (e) Protein Carbonyls. *: significant difference *vs*. SC ($P < 0.05$); **: significant difference *vs*. SC ($P < 0.01$).

endothelium-independent vasodilation between control and trained groups, also in the highest concentration of SNP (SC = 136.1%, MI = 141.6%, HI = 136.1%; $P = 0.28$), which by the way is supported by previous studies [33, 34].

Altogether, those findings support the hypothesis that the caloric cost of physical training *per se* may elicit favorable adaptations in endothelial function. Conversely, our findings did not confirm the premise that microvascular endothelial improvements would be due to greater systemic NO production, since no difference between groups was found for aortic eNOS expression. Several studies indicated that an increase in vascular eNOS would occur in response to prolonged training (>10 weeks) [2]. In the present study, the training duration of 4 weeks was perhaps too short to elicit significant changes in eNOS expression [35].

Some prior research suggested that aerobic training may improve vasoreactivity in arterioles, regardless of concomitant increasing in metabolic activity or blood flow during exercise (i.e. non-exercised muscles) [4, 32, 36]. By choosing the cremaster muscle to assess *in vivo* vasoreactivity instead of active muscle beds (soleus, gastrocnemius) as performed in most studies [2, 32, 34], we have reinforced the premise that chronic exercise effects upon microcirculation are rather systemic than local [32, 33]. This systemic response seems to be independent of exercise intensity, which gives room for interesting practical applications in regards to aerobic training and cardiovascular health.

Moreover, our findings suggest that favorable microcirculation adaptations to physical training might occur irrespective of predominance of type I fibers in a given muscle; indeed, slow oxidative fibers are often associated with higher vasodilation capacity [37], while the predominance in cremaster muscle is of fast glycolytic fibers [38]. Despite the overall strengths by deciding for cremaster muscle, the assessment of vasoreactivity solely in such muscle represents the main limitation of the present study.

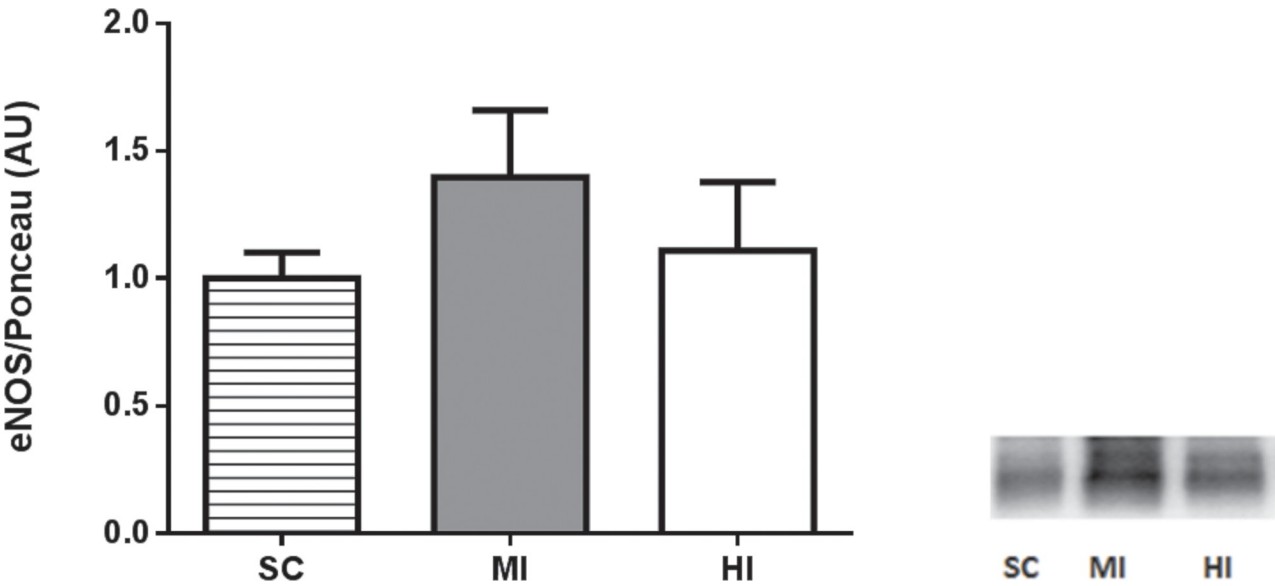

**Fig 4. Western blot analysis of aortic eNOS at the end of the experimental period (4ᵗʰ week).** Data are expressed as mean ± SEM (n = 24). No significant difference was found between the groups ($P = 0.44$).

With respect to the effects of training intensity upon myocardium, only high-intensity training improved SOD and GPx activities, while CAT did not respond to any exercise intervention ($P$ = 0.12). These findings were expected and concur with prior studies showing that activity of GPx [9, 15] and SOD [9] increased after vigorous aerobic training, while effects upon CAT remain unclear [15, 39]. The fact that MDA and protein carbonyls differ in many aspects (i.e. repair systems, chemical stability, molecular formation, or sensitivity for detection) [40, 41] can help to explain why MDA levels remained unaltered, while protein carbonyls increased. Furthermore, the increase of protein carbonyls levels in HI evokes persistent doubts in regards to whether augmentation in tissue damage by ROS-generation in the heart would be necessary to stimulate antioxidant protection, or whether this acute effect would be harmful due to constant cell-damaging [13, 14]. While this issue remains unclear, it is well elucidated that SOD plays an important protective role through superoxide dismutation [15] and GPX is capable of reducing a broad range of hydroperoxides, thereby conferring great protection against cell-damage by oxidation [7, 15].

We cannot provide a definitive explanation regarding the fact that higher exercise intensity induced greater improvements in myocardium antioxidant defenses, while gains in microvascular reactivity appeared to rely more on exercise volume than intensity. However, plausible reasons might be suggested. Firstly, it is important to consider that different protocols of exercise can elicit distinct cardiovascular adaptations, either central or peripheral. With respect to the vascular milieu, a minimal amount of aerobic exercise seems to be capable to induce changes in the endothelium via several pathways, involving both physical and chemical stimuli. The accumulated evidence suggests that the endothelium is sensitive to a minimal increase in blood flow and metabolic demands elicited by aerobic exercise, due to increased laminar shear stress, greater nitric oxide (NO) release, reduced endothelin-1 concentration, and lowered reactive oxygen species (ROS) scavenged by antioxidant enzymes [42]. The induction of SOD activity through exercise training leads to ROS detoxification, ultimately reducing the degradation of NO and improving the endothelium dependent vasodilation. In fact, several prior studies reported an improvement in endothelial function after chronic exercise performed with either high or moderate intensity *vs*. sedentary controls [31, 43].

In which concerns the adaptation to oxidative stress in the myocardium, favorable effects seems to be rather provoked by aerobic exercise performed with high- than low- to moderate intensity [9, 44]. The myocardium has an elevated mitochondrial volume and exhibits one of the highest mass-specific oxygen consumption rates in the body. Moreover, it constantly copes with high rates of oxidant formation and stress [8], which helps to explain the presence of relatively high levels of GSH antioxidants. For this reason, only high cardiac workloads are able to increase the production of superoxide and oxyradicals, thereby activating antioxidant enzymes [44]. Some evidence in rodents and humans confirms that exercise needs to be of sufficient intensity to result in accumulation of ROS and subsequent oxidative stress, which progressively improves the cardioprotective effects mediated by antioxidants [45]. Kemi et al. [46] reinforced the premise that a minimum intensity threshold would be necessary to induce myocardium adaptations, suggesting that besides requiring high intensity training, those effects would also demand several weeks to be fully active. On the other hand, improvements in endothelium-dependent vasodilation would occur earlier and in response to exercise performed with lower intensities.

Some limitations of this study must be acknowledged. Firstly, it is difficult to fully extrapolate the findings of experiments developed with animals to humans. Although biologically similar in many aspects, rats may fail to entirely mimic cardiovascular adaptations observed in humans, since they have different dynamics for the heart rate, oxygen consumption, metabolism, energy cost during exercise, or lipid metabolism [47, 48]. Moreover, antioxidant activity,

lipid peroxidation, and protein damage in vasculature have not been assessed in the present study. Those biomarkers could indicate whether exercise volume and intensity might also modulate oxidative stress in the vessels. Additionally, vasoreactivity was only assessed in the cremaster muscle. Although cremaster has been widely used for evaluating systemic vasodilation *in vivo*, its utilization limits the generalization of the present data. Hence, further studies including a translational approach are warranted to investigate the effects of training intensity upon microcirculation in other body regions.

## Conclusion

Overall, our findings corroborate the premise that a minimal increase in daily caloric expenditure promoted by exercise would be enough to maintain the endothelial health, but not to enhance antioxidants in the myocardium. Given that gains in antioxidant enzymes were only detected in the group that performed high-intensity training, it seems that vigorous exercise would be necessary to improve myocardial antioxidant defenses.

## Supporting information

**S1 Dataset. Raw data of maximal graded test, isocaloric exercise bouts, and EPOC.**
(XLSX)

**S2 Dataset. Raw data of oxidative stress, microvascular reactivity, and eNOS expression.**
(XLSX)

## Acknowledgments

The authors would like to thank Paulo José Lopes and Claudio Natalino Ribeiro for animal care, as well as João Paulo Barbosa and Lara Serrano for the support during the training of animals.

## Author Contributions

**Conceptualization:** Lorena Paes, Daniel Bottino, Eliete Bouskela, Paulo Farinatti.

**Data curation:** Lorena Paes, Daniel Lima, Cristiane Matsuura, Maria das Graças de Souza, Fátima Cyrino, Paulo Farinatti.

**Formal analysis:** Lorena Paes, Cristiane Matsuura, Maria das Graças de Souza, Fátima Cyrino, Carolina Barbosa, Fernanda Ferrão, Daniel Bottino, Eliete Bouskela, Paulo Farinatti.

**Investigation:** Lorena Paes, Daniel Lima, Cristiane Matsuura, Maria das Graças de Souza, Fátima Cyrino, Carolina Barbosa, Fernanda Ferrão, Paulo Farinatti.

**Methodology:** Lorena Paes, Daniel Lima, Maria das Graças de Souza, Fátima Cyrino, Carolina Barbosa, Fernanda Ferrão, Eliete Bouskela, Paulo Farinatti.

**Project administration:** Daniel Bottino, Eliete Bouskela, Paulo Farinatti.

**Resources:** Cristiane Matsuura, Paulo Farinatti.

**Supervision:** Cristiane Matsuura, Eliete Bouskela, Paulo Farinatti.

**Writing – original draft:** Lorena Paes, Daniel Lima, Cristiane Matsuura, Daniel Bottino, Eliete Bouskela, Paulo Farinatti.

**Writing – review & editing:** Lorena Paes, Eliete Bouskela, Paulo Farinatti.

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
