## [Decision Letter · Decision Letter 0]

27 Jun 2019

PONE-D-19-14652

Effects of moderate and high intensity isocaloric aerobic training upon microvascular reactivity and myocardial oxidative stress in rats

PLOS ONE

Dear Prof Farinatti,

Thank you for submitting your manuscript to PLOS ONE. After careful consideration, we feel that it has merit but does not fully meet PLOS ONE’s publication criteria as it currently stands. Therefore, we invite you to submit a revised version of the manuscript that addresses the points raised during the review process.

We would appreciate receiving your revised manuscript by Aug 11 2019 11:59PM. To enhance the reproducibility of your results, we recommend that if applicable you deposit your laboratory protocols in protocols.io, where a protocol can be assigned its own identifier (DOI) such that it can be cited independently in the future. For instructions see: http://journals.plos.org/plosone/s/submission-guidelines#loc-laboratory-protocols

We look forward to receiving your revised manuscript.

Kind regards,

Ioannis G. Fatouros

Academic Editor

PLOS ONE

Journal Requirements:

1.

3. Moreover, please provide the original, uncropped images for all Western blots.

Reviewers' comments:

Reviewer's Responses to Questions

**Comments to the Author**

1. Is the manuscript technically sound, and do the data support the conclusions?

Reviewer #1: Yes

Reviewer #2: Yes

2. Has the statistical analysis been performed appropriately and rigorously? 

Reviewer #1: Yes

Reviewer #2: Yes

3. Have the authors made all data underlying the findings in their manuscript fully available?

Reviewer #1: Yes

Reviewer #2: Yes

4. Is the manuscript presented in an intelligible fashion and written in standard English?

Reviewer #1: Yes

Reviewer #2: Yes

5. Review Comments to the Author

Reviewer #1: The authors present an interesting work with sufficient body of evidence. The paper is well presented and well documented. I have some major comments, though, that have to be addressed by the authors before the manuscript is accepted for publication.

1. Can the authors hypothesize why protein carbonyls increased in HI training, whereas TBARS remained unaffected? Are they aware of a putative molecular mechanism that could explain these findings given that both biomarkers refer to macromolecule oxidation?

2. Why did the authors choose a rather unusual muscle, namely cremaster, in order to evaluate microvascular reactivity?

3. Do the authors have a plausible explanation regarding the fact that higher exercise

intensity induces greater improvements in myocardium antioxidant defenses, while gains in microvascular reactivity appeared to rely more on exercise volume than intensity? Please refer to other references from the literature.

4. The English language needs improvement throughout the manuscript.

Reviewer #2: Effects of moderate and high intensity isocaloric aerobic training upon microvascular reactivity and myocardial oxidative stress in rats

The question posed by the authors of this study is interesting. Intensity of exercise and adaptations on endothelial function or myocardium antioxidant protection is an area of scientific interest. Also, studies on intensity and volume of exercise on animal models are scarce. The study is generally well written with good presentation of the data needed. I will proceed to a few suggestions to the authors.

Introduction

Line 57: “seems to be”

Methods

Line 130: A title is needed

Results/ tables/ graphs

It is not clear what the overall diet of rats is. There is data to suggest that microvascular reactivity and oxidative stress markers are influenced by diet. It would be good to mention at least the overall energy intake value (kcal/day) and the macronutrient content of the diet. Was it isocaloric and specific among the two groups of intervention? All groups gained the same kg at the end of the study; however, was quality the same? What was the diet of control group?

It is not clear whether measurements shown in graphs are taken on 4 wks. This should be written on the legends.

Line 228: “significantly lower”

Line 230: “significantly increased”

Discussion

Limitations are well pointed. Possibly it would be of interest to see some sentences regarding the speculated implications of the results on human subjects.

6. PLOS authors have the option to publish the peer review history of their article (what does this mean?). If published, this will include your full peer review and any attached files.

Reviewer #1: No

Reviewer #2: No

---

## [Author Response · Author response to Decision Letter 0]

7 Sep 2019

Reviewer #1: 

The authors present an interesting work with sufficient body of evidence. The paper is well presented and well documented. I have some major comments, though, that have to be addressed by the authors before the manuscript is accepted for publication.

1. Can the authors hypothesize why protein carbonyls increased in HI training, whereas TBARS remained unaffected? Are they aware of a putative molecular mechanism that could explain these findings given that both biomarkers refer to macromolecule oxidation?

Answer: We thank the reviewer for the opportunity to clarify this question. Some mechanisms might indeed explain the responses observed for TBARS (MDA) and protein carbonyls. One of them is the difference between the repair systems of lipid and protein oxidation. Oxidized proteins are completely degraded by proteases before protein replacement and resynthesis. In the event of high oxidative stress, the proteolytic capacity of cells may not be sufficient enough to counteract the increasing of oxidized protein molecules, which accumulate in the cells as “oxidative junk”. On the other hand, products from lipid peroxidation are easily decomposed into several reactive species (including MDA) which act as cross-linking reagents for protein oxidation (please refer to Life 2000; 50: 279-89). To summarize, MDA is less stable than protein carbonyls and its degradation to participate in protein oxidation is faster (due to a less complex system). All those features might contribute to differences of chemical stability between MDA and protein carbonyls, which may have also influenced the results. In molecular terms, oxidized protein seems to have earlier formation and higher stability than oxidized lipids. In this sense, Dalle-Done et al. (Clin Chim Acta. 2003; 329: 23-38) suggested that cells are able to degrade oxidized proteins within hours (or days), while end products of lipid peroxidation are detoxified in a few minutes (please refer to Antioxid. Redox Signal 2013;18:1208-46; and Clin Chim Acta 2003;329:23-38). Carbonyl groups have therefore earlier formation and circulate for longer time, which leads to a greater accumulation in the cells vs. MDA. This fact might increase the relative sensitivity for the detection of oxidized molecules. 

It is worthy to mention that the characteristics of the training protocol may have influenced the results regarding those biomarkers. The levels of MDA, protein carbonyl, and antioxidant enzymes are acknowledged to vary depending on the training period and assessment window (please refer to Comp Biochem Physiol 2006;143:239-45). This means that antioxidant enzymes or oxidized molecules may not equally respond to the training stimuli and at the same time. In our study, it is reasonable to speculate that albeit protein carbonyl appears altered after 4 weeks of high intensity training, it is unclear whether a longer training period would provoke similar responses. It is expected that both MDA and oxidized proteins would be lowered after 8-12 weeks as chronic adaptation to exercise training (please refer to J Appl Physiol 2000; 89: 21-8). 

In order to clarify this issue, a sentence related to these putative mechanisms were added to the Discussion section: “The fact that MDA and protein carbonyls differ in many aspects (i.e. repair systems, chemical stability, molecular formation, or sensitivity for detection) can help to explain why MDA levels remained unaltered, while protein carbonyls increased” (page 13, Lines 297-300). 

2. Why did the authors choose a rather unusual muscle, namely cremaster, in order to evaluate microvascular reactivity?

Answer: It is an important issue and we agree that our option must be justified. Actually, the effects of chronic exercise upon microvascular reactivity have been assessed in a wide range of skeletal muscles (e.g. soleus, gastrocnemius, gracilis, spinotrapezius) (please refer to Med Sci Sports Exerc 2006; 38: 445-54). Studies examining non-locomotor muscles or muscles with low- to intermediary metabolic demand are scarce. The cremaster is a striated and non-locomotor muscle without any major or direct recruitment during physical activity. Following this reasoning, we have chosen the cremaster to confirm that the effects of exercise training would be systemic rather than a consequence of increased muscle mechanical activity. 

Some prior studies have examined non-locomotor muscles to investigated the effects of exercise upon microcirculation, as the cremaster or cheek pouch (please refer to J Appl Physiol 2005;98:2113-8; J Appl Physiol Respir Environ Exerc Physiol 1981; 51: 282-7; and Microvasc Res 2014; 93: 34-41). Overall, these muscles have proven to be adequate tissues for the assessment of microcirculatory changes. Moreover, we highlight the fact that they are easily prepared due to their morphological features (i.e. thin layers, well distributed vascularization, or distinct visualization of microvessels). In short, albeit we agree with the reviewer that the cremaster is an unusual muscle to evaluate the microvascular reactivity, the assessment of its arterioles by intravital microscopy enables to visualize several biological vascular processes (please refer to Methods Mol Biol 2015; 1339: 357-66).

3. Do the authors have a plausible explanation regarding the fact that higher exercise intensity induces greater improvements in myocardium antioxidant defenses, while gains in microvascular reactivity appeared to rely more on exercise volume than intensity? Please refer to other references from the literature. 

Answer: Firstly, it is important to consider that different protocols of exercise can elicit distinct cardiovascular adaptations, either central or peripheral. With respect to the vascular milieu, a minimal amount of aerobic exercise seems to be capable to induce changes in the endothelium via several pathways, involving both physical and chemical stimuli. The accumulated evidence suggests that the endothelium is sensitive to a minimal increase in blood flow and metabolic demands elicited by aerobic exercise, due to increased laminar shear stress, greater nitric oxide (NO) release, reduced endothelin-1 concentration, and lowered reactive oxygen species (ROS) scavenged by antioxidant enzymes (please refer to Sports Med 2009; 39: 797-812). 

The induction of SOD activity through exercise training leads to ROS detoxification, ultimately reducing the degradation of NO and improving the endothelium dependent vasodilation. In fact, several prior studies reported an improvement in endothelial function after chronic exercise performed with either high or moderate intensity vs. sedentary controls (Cardio Res. 2009; 81, 723–32; Circulation. 2008; 118:346-54; J Appl Physiol. 2016; 121: 279-88). In which concerns the adaptation to oxidative stress in the myocardium, favorable effects seems to be rather provoked by aerobic exercise performed with high- than low- to moderate intensity (please refer to Am J Physiol. 1993; 265(6 Pt 2):H2094-8; Mol Med Rep. 2015; 12: 2374-82). The myocardium has an elevated mitochondrial volume and exhibits one of the highest mass-specific oxygen consumption rates in the body. Moreover, it constantly copes with high rates of oxidant formation and stress (please refer to J Appl Physiol 2000; 89: 21-8), which helps to explain the presence of relatively high levels of GSH antioxidants. For this reason, only high cardiac workloads are able to increase the production of superoxide and oxyradicals, thereby activating antioxidant enzymes (please refer to Mol Med Rep. 2015; 12: 2374-82). Some evidence in rodents and humans confirms that exercise needs to be of sufficient intensity to result in accumulation of ROS and subsequent oxidative stress, which progressively improves the cardioprotective effects mediated by antioxidants (please refer to Adv Clinic Chem 2008; 46: 1-50). Kemi et al. (Cardio Res 2005; 67: 161-72) reinforced the premise that a minimum intensity threshold would be necessary to induce myocardium adaptations to oxidative stress, suggesting that besides requiring high intensity training, those effects would also demand several weeks to be fully active. On the other hand, improvements in endothelium-dependent vasodilation would occur earlier and in response to exercise performed with lower intensities. All this rationale was added to the Discussion section (Pages 14-15, Lines 307-336).

4. The English language needs improvement throughout the manuscript.

Answer: We double-checked the manuscript for spelling and grammar flaws.

Reviewer #2: Effects of moderate and high intensity isocaloric aerobic training upon microvascular reactivity and myocardial oxidative stress in rats.

The question posed by the authors of this study is interesting. Intensity of exercise and adaptations on endothelial function or myocardium antioxidant protection is an area of scientific interest. Also, studies on intensity and volume of exercise on animal models are scarce. The study is generally well written with good presentation of the data needed. I will proceed to a few suggestions to the authors.

Introduction

1) Line 57: “seems to be”

Answer: The correction was made.

Methods

2) Line 130: A title is needed

Answer: The correction was made

Results/ tables/ graphs

3) It is not clear what the overall diet of rats is. There is data to suggest that microvascular reactivity and oxidative stress markers are influenced by diet. It would be good to mention at least the overall energy intake value (kcal/day) and the macronutrient content of the diet. Was it isocaloric and specific among the two groups of intervention? All groups gained the same kg at the end of the study; however, was quality the same? What was the diet of control group?

Answer: We thank the reviewer for the opportunity to clarify this question. Indeed, there are several evidences demonstrating that diet composition can affect oxidative stress (please refer to Int J Vitam Nutr Res. 2001; 71:339-46; Nutr J 2011; 10: 122; Redox Biol 2016; 8: 216-25). High-fat diets, for example, are able to increase the efflux of non-esterified fatty acids, the production of ROS and the levels of lipid peroxidation, resulting in oxidative imbalance and endothelial cell impairment. In the present experiment, only a manufactured standard rat chow (Nuvilab CR-1, NuvitalTM) was offered to animals assigned in SC, HI and MI. Considering that this standard chow provided 3.78 kcal/g (56% carbohydrates, 22% proteins and 4% lipids), it is possible to claim that all groups were kept in isocaloric diet conditions. In order to elucidate this issue, we have included in the Results section data of body mass evolution (new Figure 1) and overall energy intake (kcal/day) (Page 9, Lines 206-208). 

4) It is not clear whether measurements shown in graphs are taken on 4 wks. This should be written on the legends.

Answer: We added this information in the legends, as follows (Page 23, Lines 591-605): 

“Fig. 1 – Body mass during the experimental period (4 weeks). Data are expressed as mean ± SEM (n = 24). No significant difference was found between the groups (P = 0.59).

Fig. 2 − Microvascular reactivity in vivo of arterioles in the cremaster muscle at the end of the experimental period (4th week). Data are expressed as median (1st − 3rd quartile) (n = 24). (a) Endothelium-dependent vasodilation; (b) endothelium-independent vasodilation. * Significant difference vs. SC (P < 0.05).

Fig. 3 – Biomarkers of oxidative stress analyzed on left ventricle samples at the end of the experimental period (4th week). Data are expressed as mean ± SEM (n = 24). (a) CAT Activity; (b) SOD Activity; (c) GPX Activity; (d) TBARS - MDA content; (e) Protein Carbonyls. *: significant difference vs. SC (P < 0.05); **: significant difference vs. SC (P < 0.01). 

Fig. 4 – Western blot analysis of aortic eNOS at the end of the experimental period (4th week). Data are expressed as mean ± SEM (n = 24). No significant difference was found between the groups (P = 0.44)”. 

5) Line 228: “significantly lower”

Answer: The correction was made.

6) Line 230: “significantly increased”

Answer: The correction was made.

Discussion

7) Limitations are well pointed. Possibly it would be of interest to see some sentences regarding the speculated implications of the results on human subjects.

Answer: This suggestion is pertinent. Even though transferring the meaning of data obtained in rodents to humans is always to a certain degree speculative, we can at least suggest potential implications. Mice and rats can share up to 98% of DNA with humans – as a result, in appropriate experimental conditions these animals may exhibit similar cardiovascular problems that afflict human beings, as well as benefit of specific interventions (e.g. pharmacological treatment, exercise, or diet). In contrast to humans, research with rodents allows investigating in detail the isolate effects of those interventions upon blood vessels and cardiac tissues (please refer to Biomed Res Int. 2015; 2015: 528757). On the other hand, there are many physiological differences between rats and humans that must be considered when extrapolating the results of experiments with rodents to humans (i.e., fiber types, speed of muscle contraction, muscle metabolic activity, energy cost of exercise, heart and respiratory rate, oxygen consumption, and lipid metabolism) (please refer to Physiol Rep. 2015; 3(2): e12293). In order to address this particular issue, we have included some sentences in the limitation’s paragraph, as follows: “Firstly, it is difficult to fully extrapolate the findings of experiments developed with animals to humans. Although biologically similar in many aspects, rats may fail to entirely mimic cardiovascular adaptations observed in humans, since they have different dynamics for the heart rate, oxygen consumption, metabolism, energy cost during exercise, or lipid metabolism (Biomed Res Int 2015; 2015: 528757; Physiol Rep 2015; 3: e12293)” (Page 15, Lines 337-341).

REFERENCES

Boa, B.; Costa, R.; Souza, M. et al. Aerobic exercise improves microvascular dysfunction in fructose fed hamsters. Microvasc Res. 2014; 93: 34-41.

Bloomer, R.; Effect of exercise on oxidative stress biomarkers. Adv Clinic Chem. 2008; 46:1-50. 

Davies, K. Oxidative Stress, Antioxidant Defenses, and Damage Removal, Repair, and Replacement Systems. Life. 2000; 50: 279–89. 

Dalle-Done, I.; Rossi, R.; Giustarini, D. et al. Protein carbonyl groups as biomarkers of oxidative stress. Clin Chim Acta. 2003; 329: 23–38. 

Francescomarino, S.; Sciartilli, A.; Valerio, V. et al. The Effect of Physical Exercise on Endothelial Function. Sports Med. 2009; 39: 797-812.

Georgios Goutianos, G.; Tzioura, A.; Kyparos, A. et al. The rat adequately reflects human responses to exercise in blood biochemical profile: a comparative study. Physiol Rep. 2015; 3(2): e12293.

Gul, M.; Demircan, B.; Taysi, S. et al. Effects of endurance training and acute exhaustive exercise on antioxidant defense mechanisms in rat heart. Comp Biochem Physiol. 2006, 143: 239-45. 

Haram, P.; Kemi, O.; Lee, S. et al. Aerobic interval training vs. continuous moderate exercise in the metabolic syndrome of rats artificially selected for low aerobic capacity. Cardio Res. 2009; 81, 723–32. 

Harris, P.; Joshua, I.; Miller, F. Decreased vascular sensitivity to norepinephrine following exercise training. J Appl Physiol Respir Environ Exerc Physiol. 1981; 51: 282-7.

Jasperse, J.; Laughlin, H. Endothelial function and exercise training: Evidence from studies using animal models. Med Sci Sports Exerc. 2006; 38: 445-54.

Kakimoto, P.; Kowaltowski, A. Effects of high fat diets on rodent liver bioenergetics and oxidative imbalance. Redox Biol 2016; 8: 216–25. 

Kemi, O.; Haram, P. Loennechen, J. et al. Moderate vs. high exercise intensity: Differential effects on aerobic fitness, cardiomyocyte contractility, and endothelial function. Cardio Res. 2005; 67: 161-72. 

Leong, X.; Ng, C.; Jaarin, K. Animal Models in Cardiovascular Research: Hypertension and Atherosclerosis. Biomed Res Int. 2015; 2015: 528757.

Liu, J.; Yeo, H.; O¨Vervik-Douki, E.; et al. Chronically and acutely exercised rats: biomarkers of oxidative stress and endogenous antioxidants. J Appl Physiol. 2000; 89: 21–8.

Lu, Y.; Chiang, C. Effect of Dietary Cholesterol and Fat Levels on Lipid Peroxidation and the Activities of Antioxidant Enzymes in Rats. Int J Vitam Nutr Res. 2001; 71: 339-46. 

Lu, K.; Wang, L. Wang, C. et al. Effects of high-intensity interval versus continuous moderate‑intensity aerobic exercise on apoptosis, oxidative stress and metabolism of the infarcted myocardium in a rat model. Mol Med Rep. 2015; 12: 2374-82.

Orth, T.; Allen, J.; Wood, J. et al. Exercise training prevents the inflammatory response to hypoxia in cremaster venules. J Appl Physiol. 2005; 98: 2113-8. 

Peairs, A.; Rankin, J.; Lee, Y. Effects of acute ingestion of different fats on oxidative stress and inflammation in overweight and obese adults. Nutr J. 2011; 10: 122. 

Powers, S.; Criswell, D.; Lawler, J. et al. Rigorous exercise training increases superoxide dismutase activity in ventricular myocardium. Am J Physiol. 1993; 265(6 Pt 2): H2094-8.

Radak, Z.; Zhao, Z.; Koltai, E. et al. Oxygen Consumption and Usage During Physical Exercise: The Balance Between Oxidative Stress and ROS-Dependent Adaptive Signaling. Antioxid. Redox Signal. 2013; 18: 1208-46. 

Rius, C.; Sanz, M. Intravital Microscopy in the Cremaster Muscle Microcirculation for Endothelial Dysfunction Studies. Methods Mol Biol. 2015; 1339: 357-66. 

Tjønna, A.; Lee, S.; Rognmo, Ø. et al. Aerobic interval training versus continuous moderate exercise as a treatment for the metabolic syndrome: a pilot study. Circulation. 2008; 118: 346-54.

Sawyer, B.; Tucker, W.; Bhammar, D. et al. Effects of high-intensity interval training and moderate-intensity continuous training on endothelial function and cardiometabolic risk markers in obese adults. J Appl Physiol. 2016; 121: 279-88.

---

## [Editor Report · Decision Letter 1]

28 Oct 2019

PONE-D-19-14652R1

Effects of moderate and high intensity isocaloric aerobic training upon microvascular reactivity and myocardial oxidative stress in rats

PLOS ONE

Dear Prof Farinatti,

Thank you for submitting your manuscript to PLOS ONE. After careful consideration, we feel that it has merit but does not fully meet PLOS ONE’s publication criteria as it currently stands. Therefore, we invite you to submit a revised version of the manuscript that addresses the points raised during the review process. Please address all comments raised by the reviewers as well as the following comments of the editor:

1. Please label appropriately  the western blot images provided. I have examined the revised paper as well as the figure in question and the raw data. The bands of fig. 4 are very blurry as they are in the original blot. As it seems, authors actually measured the protein levels of three different groups of rats following the intervention. There is no ladder or control in the original figure. The original figure has 9 columns that I am not sure what they represent since there are no labels. Each column must represent a different animal from one of the three experimental groups. Authors need to provide the original membranes from all the animals (24 in total) and label each column to let us know which column represents a specific animal of a specific group. Therefore, I invite you to provide the original membranes from all the animals (ideally, a control and ladder should also be provided). 

2. Please explain why the study design did not include a  measurement of the dependent variables at baseline, to allow the evaluation  of delta differences between pre- and post-training values in the three groups. 

We would appreciate receiving your revised manuscript by Nov 28 2019 11:59PM. To enhance the reproducibility of your results, we recommend that if applicable you deposit your laboratory protocols in protocols.io, where a protocol can be assigned its own identifier (DOI) such that it can be cited independently in the future. For instructions see: http://journals.plos.org/plosone/s/submission-guidelines#loc-laboratory-protocols

We look forward to receiving your revised manuscript.

Kind regards,

Ioannis G. Fatouros

Academic Editor

PLOS ONE

---

## [Author Response · Author response to Decision Letter 1]

19 Nov 2019

Rio de Janeiro, November 14th, 2019

- Code: PONE-D-19-14652R1

- Title: “Effects of moderate and high intensity isocaloric aerobic training upon microvascular reactivity and myocardial oxidative stress in rats"

- Corresponding Author: Paulo Farinatti 

- e-mail: ptvf1964@gmail.com

Dear Editor,

Please find below our responses to the reviewers’ comments concerning our manuscript (PONE-D-19-14652R1), entitled “Effects of moderate and high intensity isocaloric aerobic training upon microvascular reactivity and myocardial oxidative stress in rats”. Thank you for the opportunity of resubmitting our manuscript. 

We have addressed all the issues raised by the reviewers. The manuscript has been rewritten according to the reviewers’ suggestions, and an itemized, point-by-point response to each of the reviewers’ comments has been provided. Changes on the manuscript are marked in the “Revised Manuscript with Track Changes”, as per the received instructions.

Yours Sincerely,

Paulo Farinatti, PhD

University of Rio de Janeiro State

Salgado de Oliveira University

 

1) Please label appropriately the western blot images provided I have examined the revised paper as well as the figure in question and the raw data. The bands of fig. 4 are very blurry as they are in the original blot. As it seems, authors actually measured the protein levels of three different groups of rats following the intervention. There is no ladder or control in the original figure. The original figure has 9 columns that I am not sure what they represent since there are no labels. Each column must represent a different animal from one of the three experimental groups. Authors need to provide the original membranes from all the animals (24 in total) and label each column to let us know which column represents a specific animal of a specific group. Therefore, I invite you to provide the original membranes from all the animals (ideally, a control and ladder should also be provided).

Answer: We have sent the figures with the molecular weight and labels for each group. Below the pictures we placed the overlay of blot images with the colorimetric image showing the weight markers. We hope that this information will be enough to satisfy the Editor’s demand.

2) Please explain why the study design did not include a measurement of the dependent variables at baseline, to allow the evaluation of delta differences between pre- and post-training values in the three groups

Answer: Measurements of some outcomes at baseline were not feasible, since animals should be sacrificed. Alternatively, we could have added three more groups of rats to be sacrificed in control conditions (surgical procedures to expose cremaster muscle, and extraction of aorta and left ventricle). However, the value of this “control assessment” would be limited, since pre vs. post-training comparisons would not be inter-individual. The refinement of the experiment by doing so would not be questionable and the unnecessary sacrifice of animals would violate the ethical requisite of 3R's (Replacement, Reduction and Refinement) (Please refer to Bate S, Karp N. PLoS One 2014; 9: e114872). Given these ethical reasons, repeated measurements to evaluate deltas were performed only for variables which assessment was not invasive, and when surgery procedures were not necessary (e.g. VO2 max, body weight). In the case of variables invasively assessed, instead of performing repeated measurements we adopted an approach widely used in prior research with animals, by including only one control group (sedentary animals) that directly contrasted with both experimental groups (exercised animals) (Please refer to Johnson P, Besselsen D. ILAR Journal, 2002; 43: 202-6). In fact, this procedure is usually applied in studies investigating cardiovascular outcomes, particularly microcirculation (Please refer to Machado M et al. Exp Physiol, 2017 102: 1716–28; Boa B et al. PLoS ONE 2014, 9: e102554). Finally, we could claim that outcomes in our study have not been affected by potential confounding variables (as aging, diet, or exercise protocol). Thus, a single control group seemed to be enough to provide data corresponding to “baseline condition” (i.e. without any intervention).

---

## [Decision Letter · Decision Letter 2]

24 Jan 2020

Effects of moderate and high intensity isocaloric aerobic training upon microvascular reactivity and myocardial oxidative stress in rats

PONE-D-19-14652R2

Dear Dr. Farinatti,

We are pleased to inform you that your manuscript has been judged scientifically suitable for publication and will be formally accepted for publication once it complies with all outstanding technical requirements.

With kind regards,

Ioannis G. Fatouros

Academic Editor

PLOS ONE

Additional Editor Comments (optional):

Reviewers' comments:

Reviewer's Responses to Questions

**Comments to the Author**

1. If the authors have adequately addressed your comments raised in a previous round of review and you feel that this manuscript is now acceptable for publication, you may indicate that here to bypass the “Comments to the Author” section, enter your conflict of interest statement in the “Confidential to Editor” section, and submit your "Accept" recommendation.

Reviewer #1: All comments have been addressed

Reviewer #2: All comments have been addressed

2. Is the manuscript technically sound, and do the data support the conclusions?

Reviewer #1: Yes

Reviewer #2: Yes

3. Has the statistical analysis been performed appropriately and rigorously? 

Reviewer #1: Yes

Reviewer #2: Yes

4. Have the authors made all data underlying the findings in their manuscript fully available?

Reviewer #1: Yes

Reviewer #2: Yes

5. Is the manuscript presented in an intelligible fashion and written in standard English?

Reviewer #1: Yes

Reviewer #2: Yes

6. Review Comments to the Author

Reviewer #1: The authors have successfully addressed my comments in the revised version of the manuscript. Therefore, I recommend the publication of the manuscript.

Reviewer #2: Comments have been addressed and authors made changes in the text. Some details missing are now added (eg time points of the measurements are now added in figure legends). Also explanation in answers is backed with literature.

7. PLOS authors have the option to publish the peer review history of their article (what does this mean?). If published, this will include your full peer review and any attached files.

Reviewer #1: No

Reviewer #2: No

---

## [Editor Report · Acceptance letter]

31 Jan 2020

PONE-D-19-14652R2 

Effects of moderate and high intensity isocaloric aerobic training upon microvascular reactivity and myocardial oxidative stress in rats 

Dear Dr. Farinatti:

I am pleased to inform you that your manuscript has been deemed suitable for publication in PLOS ONE. Congratulations! Your manuscript is now with our production department. 

With kind regards,

on behalf of

Dr. Ioannis G. Fatouros 

Academic Editor

PLOS ONE